# Effects of paediatric schistosomiasis control programmes in sub-Saharan Africa: A systematic review

**Maryline Vere** [1]*, **Wilma ten Ham-Baloyi**[1,2], **Paula Ezinne Melariri**[2]

1 Faculty of Health Sciences, Department of Environmental Health, Nelson Mandela University, University Way, Summerstrand, Gqeberha, South Africa, 2 Faculty of Health Sciences, Department of Nursing Science, Nelson Mandela University, University Way, Summerstrand, Gqeberha, South Africa

☯ These authors contributed equally to this work.

* s226225283@mandela.ac.za

**Data Availability Statement:** All relevant data are within the paper and its Supporting Information files.

## Abstract

Preventive chemotherapy by mass drug administration is globally recommended as the primary method of reaching the elimination of schistosomiasis, especially in the high risk-paediatric population. This systematic review provides a summary of the effects of paediatric schistosomiasis control programs on eliminating schistosomiasis in sub-Saharan Africa. A systematic search was conducted in PubMed, EBSCOhost, and other databases to obtain studies regarding the effects of paediatric schistosomiasis control programmes in sub-Saharan Africa. 3455 studies were screened for eligibility, included articles reported on both paediatrics control programmes and schistosomiasis, and articles were excluded when they did not report on schistosomiasis control programmes in paediatrics exclusively. 40 selected studies were critically appraised using the JBI critical appraisal tools for relevance and 30 studies were included in the study. An in-depth quantitative descriptive analysis was conducted, and a comprehensive narrative summary explained the results within the scope of the review questions. The results show that despite preventive chemotherapy lowering schistosomiasis prevalence, chances of re-infection are high in endemic areas. Preventive chemotherapy without complementary interventions including safe water provision and proper sanitation, snail control and health education on the aetiology of schistosomiasis, transmission pattern and control practices might not eliminate schistosomiasis.

## Introduction

Schistosomiasis also called bilharzia, is a neglected tropical disease (NTD) which has remained a public health problem in Africa, Asia, and South America [1]. It is a waterborne illness transmitted by freshwater snails (*Bulinus and Biomphalaria species*). *Schistosoma* trematode worms live in intravascular regions of the urogenital tract and intestines. Globally, 732 million people are susceptible to infection with over 200 million individuals infected [2]. Most infected people live in poorly resourced communities without access to safe water for domestic use, fishing, and agriculture in the tropic and subtropic regions [3]. *Schistosoma haematobium* (*S.*

**Funding:** The author(s) received no specific funding for this work.

**Competing interests:** The authors have declared that no competing interests exist.

**Abbreviations:** NTD, Neglected tropical diseases; PSAC, pre-school aged children; SAC, school-aged children; MDA, mass drug administration; WASH, water sanitation hygiene; WHO, World Health Organization; DNA, deoxyribonucleic acid; PCR, polymerase chain reaction; ELISA, enzyme linked immunosorbent assay; CDC, Centre for Disease Control; PZQ, praziquantel.

*haematobium*) causes urogenital tract infections whereas intestinal infections are caused by *Schistosoma mansoni (S. mansoni)*, *Schistosoma intercalatum*, *Schistosoma mekongi*, *Schistosoma japonicum* and *Schistosoma guineensis*. In sub-Saharan Africa, schistosomiasis is caused by two strains, *S. haematobium and S. mansoni* [4]. The symptoms of the acute form of the infection include Katayama fever, rash, diarrhoea, nausea, and stomach pains. If left untreated individuals may develop the chronic form of the disease which includes bladder cancer, prostate cancer, spleen and liver enlargement, infertility, urogenital tract ulceration and kidney blockage [5]. In schistosomiasis endemic areas there is decent exposure in pre-school aged children (PSAC) (≤5 years), despite the age group not participating in most schistosomiasis mass control programmes in the past years due to a lack of data on the safety and specificity of praziquantel dosage for this specific age group [6]. PSAC become infected during visits to the water bodies as they go with their caregivers or older siblings, hence the exposure is passive. They are infected by schistosomal cercariae in the banks of waterbodies where the Schistosoma vectors *Bulinus* and *Biomphalaria* snails are mostly found, whilst waiting for their caregivers perform houseold chores [5,7]. School-aged children (SAC) have the highest prevalence of schistosomiasis infections compared to other age groups due to their frequent contact with infested water bodies, this age group is considered an at high-risk group for schistosomal infections [2]. Lack of good hygiene practices and proper sanitation, paediatrics have a higher risk of exposure to schistosomal parasites because they swim, bath, and fish in infested water frequently than other age groups.

Schistosomiasis control, monitoring, and elimination programs have been implemented in most of the endemic countries in sub-Saharan Africa. Without improvements in environmental conditions, reinfection can occur after treatment, necessitating periodic administration of praziquantel, once every one or two years, depending on prevalence rates [8]. Transmission is greatly reduced by interventions such as introducing washing sinks, swimming pools, and providing domestic water supplies such as piped water, tap water and fences along water bodies. Although in some settings activities like fishing, sand harvesting and agriculture could still expose people to infections. With 75% coverage, three years of annual treatment of SAC with praziquantel could achieve control of morbidity, and continuation of mass drug administration (MDA), for five to six more years will have a higher probability to reach the end of transmission [9]. Combining preventive chemotherapy, health education, WASH programs and snail control will facilitate reaching the end of transmission which is when the infection rate reaches zero [10]. WHO recommends preventive chemotherapy and test-and-treat approaches for PSAC and SAC, along with health education, improved sanitation, and water access, however, no studies have been done to assess the collective effects of the schistosomiasis control programs in sub-Saharan Africa on the paediatric population [11]. Systematic reviews done in the field have reported on the impact of targeted treatment and mass drug administration delivery strategies, but no further exploration was done on different intervention strategies [4]. Sokolow et al. [2016] evaluated the effectiveness of different schistosomiasis control strategies used globally over the 20th century. However, the study only includes historical information on control measures which mostly targeted the snail intermediate hosts as the main intervention strategy. The development of paediatric-friendly dosages of praziquantel that revolutionised control of schistosomiasis in the 21st century was not discussed in the previous study [12]. Hence the current systematic review, assessed the effects of paediatric schistosomiasis control programs in sub-Saharan Africa. Different control programs which include MDA/ preventive chemotherapy to reduce disease through periodic large-scale treatment with praziquantel, WASH, snail control and health education for behavioural modification were explored. The review data will provide an overview of the different paediatric schistosomiasis control

programmes implemented in sub-Saharan Africa and their success in controlling schistosomiasis in terms of eliminating schistosomiasis in the paediatric population.

## Materials and methods

### Search strategy

A comprehensive systematic literature search relevant to the topic and review objectives was conducted following the Preferred Reporting Items for Systematic Reviews and Meta-Analyses (PRISMA) guidelines [13]. The protocol of the current review was developed prospectively and registered on the online database Prospective Register of Systematic Reviews (PROSPERO) with registration number: CRD42023401938. Based on the following review question: "What are the effects of paediatric schistosomiasis control programmes the elimination of schistosomiasis in sub-Saharan Africa?", a search was conducted from March to April 2023 using five electronic databases (MEDLINE Complete, PubMed, CINAHL, Embase, Cochrane Library), followed by a manual search done for two grey literature pre-print savers (MedRexiv and BioRexiv) and a citation search -ensuring a comprehensive search - from January 2000 to March 2023 for eligible studies. Secondarily sourced articles were identified in a process of checking references in articles identified through the search and assessing them for inclusion in the review. The following search terms were used: "schistosomiasis OR bilharzia AND "preventive OR chemotherapy OR control OR elimination OR wash OR sanitation OR hygiene OR education OR mass drug administration OR snail control OR molluscicides.

### Eligibility criteria

Peer-reviewed articles reporting on paediatric schistosomiasis control programs in sub-Saharan Africa were included. Included articles matched the following criteria: (i) reported on both paediatrics control programmes in terms of reducing and eliminating schistosomiasis (ii) population studies (articles with a defined sample size and age from 1 to 18 years used as a limiter in database searches) on the co-existence of schistosomiasis and its control in sub-Saharan Africa (iii) longitudinal intervention, cluster randomized trial, prospective cohort, repeated cross sectional studies, longitudinal pre- and post-test, case control and interventive prospective studies (iv) published in peer-reviewed journal and (v) in English. Articles were excluded if they (i) did not report on schistosomiasis control programmes in paediatrics exclusively, (ii) reported on just paediatric schistosomiasis only, (iii) reported on just schistosomiasis control programmes only, or (v) they were reviews only (vi) praziquantel efficacy only (v) co-infection analysis with soil-transmitted helminthiasis and malaria, these were applied as limits to the database interface during the initial search.

### Selection process

Zotero (version 7) software was utilised as a reference manager to retrieve articles from the selected databases. Duplicates were removed and the remaining articles were exported to an Excel spreadsheet. Articles were initially selected by title screening; the articles were then screened by abstracts and the selected articles were screened by full text. Microsoft Excel spreadsheets were used throughout the article selection process. The selection was done by the first author (MV) and verified by the second and third authors (WtHB and PEM).

### Critical appraisal and risk of bias

Quality appraisal of the relevant articles was assessed by the first author and two reviewers independently, using the Joanna Briggs Institute (JBI) (*JBI*, n.d.) critical appraisal tool for the

assessment of prevalence studies. The total quality score was computed for each study using the JBI Critical Appraisal Checklist for studies reporting prevalence data with answers scored as 'No' or 'Yes' or Unclear or N/A (not applicable), respectively. Total risk scores varied between 0 and 100% where 0–50% = (High risk); 50–70% = (Moderate risk) and 71–100% = (Low risk). The total quality scores of articles included in this review were moderate to high across studies. In this review, 30 studies were critically appraised and included in the analysis.

### Data extraction and analysis

Articles were analysed in detail by MV who initially screened all articles retrieved from the electronic databases by title and the results were verified by two reviewers (WtHB and PEM). Studies that did not meet the requirements stipulated in the inclusion criteria were excluded while articles that were aligned in answering the proposed review questions were retrieved for full assessment. Any uncertainties in inclusion would be resolved through a consensus discussion by the two reviewers (WtHB and PE). Finally, the first author (MV) extracted the following information from the eligible studies included in the systematic review:

1. Study profile (authors, year of publication, country),

2. Study design and aims.

3. Targeted population (ranging from 1–18 years)

4. Sample size.

5. Type of control strategy (WASH, MDA or preventive chemotherapy, health education, behavior change and snail control)

6. Diagnosis technique

7. School-based or community-based intervention

Data from the results in the included studies were tabulated into Excel spreadsheets and in-depth quantitative descriptive analysis was conducted to sort and visualize the data into tables and bar graphs. Within the parameters of the review objective, a thorough narrative summary was created to explain the graphs and tabulated data.

## Results

### Search and selection results

The current systematic review clustered studies with critical emphasis on the effects of schistosomiasis control programmes in sub-Saharan Africa. A sum of 5445 articles was identified from five electronic databases. 1990 were screened by title and duplicate articles were removed leaving 3455 studies eligible for screening by abstract. A total of 3257 irrelevant articles were excluded during screening and 198 studies were sorted for retrieval for full assessment. No additional eligible articles were identified through the manual search of two grey literature pre-print savers (Medrix and Biorexiv) and the citation search. Finally, 30 studies were eligible for the review after 168 studies were excluded for not meeting the study inclusion criteria. The predominant reasons for the exclusion of the studies were due to the reports on praziquantel efficacy (n = 84), co-infection with soil-transmitted helminthiasis and malaria (n = 43) and no interventions included (n = 41). The PRISMA flow diagram for the study's technique of identifying, filtering, and including studies is displayed in Fig 1.

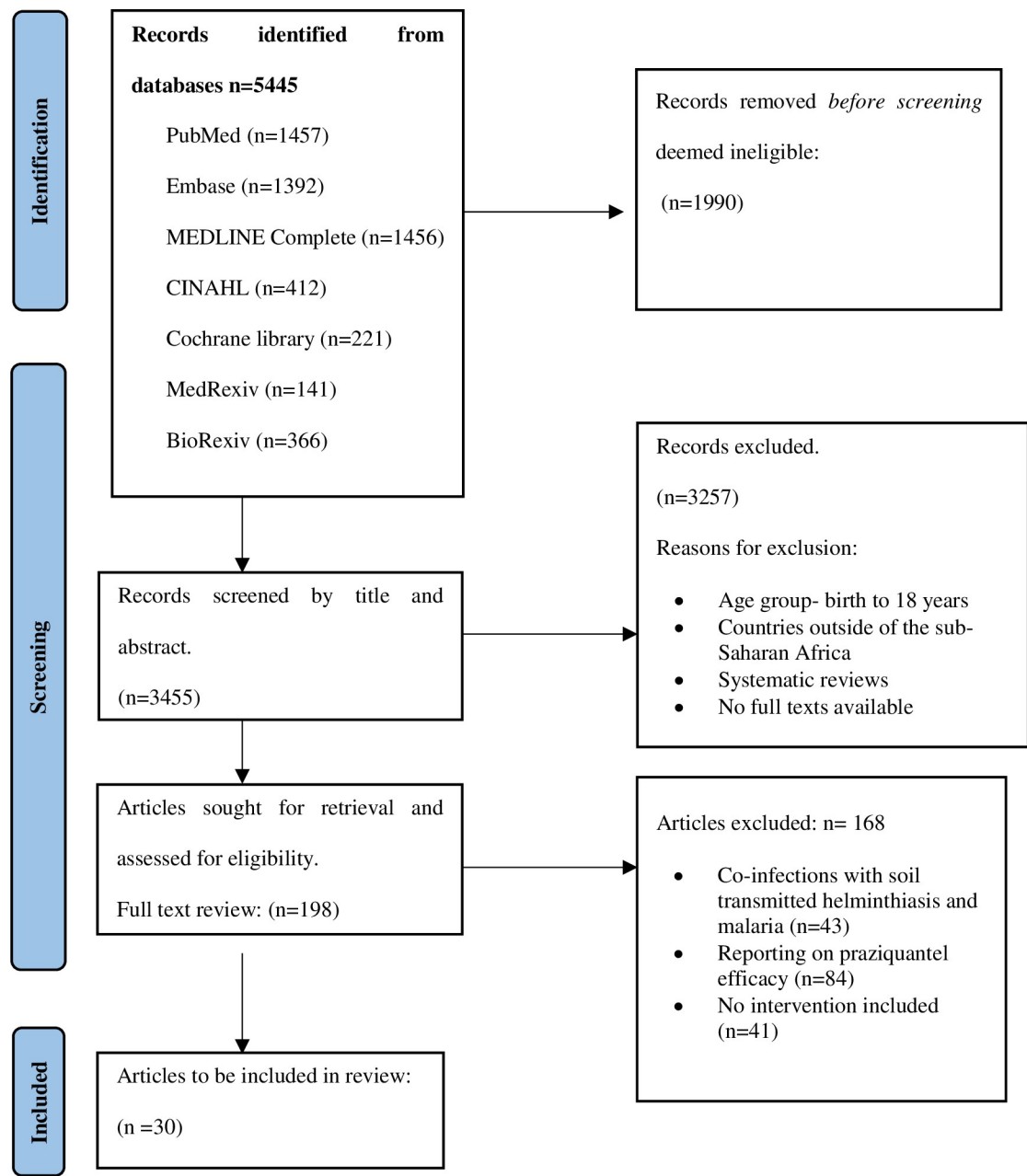

**Fig 1. PRISMA flow diagram.** Illustration of search and article selection [14].

## Study characteristics

Table 1 summarizes all of the studies considered in this systematic review. A total of 30 sub-Saharan African articles were included in the final analysis. The studies were from the following 11 countries: the majority of research (five (n = 5) studies were done in Kenya and Tanzania, followed by Nigeria and Cote d'Ivoire (n = 4), Zimbabwe and Ethiopia (n = 3), Senegal (n = 2) and 1 study conducted in each of the following countries, Togo, Angola, Sudan and Madagascar. Concerning age groups, 17 studies were predominantly on SAC only, 8 studies on both SAC and adolescents; 3 studies on PSAC only [6,15,16] and 1 study on both PSAC

**Table 1. Studies included in the review.** Summary of the studies used to evaluate the effect of paediatric schistosomiasis control programmes in sub-Saharan Africa.

| SN | First author | Country | Sample size | Age group (Years) | Control strategy | Study Design | *Schistosoma* species | Diagnosis technique | Intervention (community or school-based or combined) | Summary of main findings Principal findings |
|---|---|---|---|---|---|---|---|---|---|---|
| 1 | [25] | Zimbabwe | 212 105 boys 107 girls | SAC 7–13 years | MDA 40 mg/kg | Longitudinal intervention study | *S. haematobium S. mansoni* | Urine filtration Kato Katz ELISA | School-based | *S. haematobium* prevalence changed from 23.1% to 0.47%. Significant reduction in prevalence, intensity and re-infection levels after treatment. Biennial treatment kept infection levels low, chemotherapy also led to decreased levels of Ig G4 and Ig G1 |
| 2 | [26] | Kenya | 7500 SAC from 75 schools Cohort | SAC 9-12years | MDA 40 mg/kg | Cluster randomized trial | *S. mansoni* | Kato Katz | School-based | Considerable reductions in the annual prevalence and severity of infection in the treated arms. There was no discernible difference in the reduction of infection prevalence or intensities between Arm 1 (which received four treatments over a five-year period), Arm 2 (which received yearly treatments for five years), and Arm 3 (which received biennial treatments). In locations with moderate prevalence, biennial and annual therapy may have comparable advantages, allowing MDA to be administered to twice as many schools with the same resources as yearly MDA. |
| 3 | [27] | Kenya | 100 individuals from 25 villages: 6 arms with 25 villages each Cohort | SAC 9–12 years | MDA 40 mg/kg | Cluster randomized trial | *S. mansoni* | Kato Katz | Community-based School-based | School based treatment had higher coverage than community based treatment even after household to household treatment. School based treatment had >89% coverage whilst community based treatment had <80% coverage. Annual and biennial treatments had the same effects on the total prevalence. The prevalence decreased significantly in year 5 than in year one, Infection intensity decreased significantly. |
| 4 | [28] | Tanzania | 14620 SAC | SAC 13–14 years | MDA 40 mg/kg | Cluster randomized trial | *S. mansoni* | Kato Katz | Community-based School-based | Compared to four times school-based therapy, community-wide treatment did not result in a significantly decreased prevalence or intensities of infection in school-aged children (9 to 12 years old). The results of annual and biennial treatments were same. Unexpectedly, there was a notable rise in mean prevalence from years 4 to 5, which may have been caused by increased transmission after the year 4 MDA and the El Nino rains at that time. 79% of the community had praziquantel treatment, and the prevalence dropped from 54.6% at baseline to 45.2% in year five. |

*(Continued)*

**Table 1.** (Continued)

| SN | First author | Country | Sample size | Age group (Years) | Control strategy | Study Design | *Schistosoma* species | Diagnosis technique | Intervention (community or school-based or combined) | Summary of main findings Principal findings |
|---|---|---|---|---|---|---|---|---|---|---|
| 5 | [29] | Cote d'Ivoire | 111 individuals 60 females 50 males | SAC, Adolescents 7–15 years | MDA 40 mg/kg | Cross sectional study | *S. mansoni* | Kato Katz PCR | School-based | Detection of cell-free schistosomal DNA via PCR was more sensitive than parasitological methods used, KK, haematuria and urine filtration. After MDA, diagnosis using haematuria and egg detection missed the detection of *S. haematobium* infections but PCR detected 11% infections and 8% *S. mansoni* infections were Kato Katz had not detected schistosomiasis ova. It has been demonstrated that low-intensity infections, particularly those that follow MDA, make it difficult to detect parasite eggs in the urine and stool. |
| 6 | [30] | Zimbabwe | 27 cases 106 controls | PSAC | MDA 40 mg/kg | Case-control study | *S. mansoni* *S. haematobium* | Urine filtration Kato Katz | Community-based | The prevalence was 22% at baseline and dropped to 3.6% six months after treatment as a result of post-treatment reinfections. The Foundations of Learning domain performance of PSAC with *S. haematobium* infections was found to be 3.9 times lower than that of uninfected PSAC (p = 0.008); however, following MDA, improvements were observed in the Language and Communication Domain, Eye-Hand Coordination Domain, and General Development Domain, highlighting the significance and beneficial effects of treating schistosomiasis in PSAC. |
| 7 | [19] | Ethiopia | 80475 children, sex-matched, from 1645 schools | SAC, Adolescents (10-15yrs) | WASH | - | *S. mansoni* | Kato Katz | School-based | When comparing *S. mansoni* infection and sanitation, no discernible changes were found. Enhancing school WASH could lower the spread of schistosomal infections. With water and S. mansoni showing the strongest correlation, different WASH approaches seemed to have varying effects on infection with the various parasites. |
| 8 | [31] | Ethiopia | 8002 children Infected: 408 Uninfected:7594 | SAC (5-14yrs) | MDA 40 mg/kg | Cross sectional study | *S. mansoni* | Kato Katz | Community-based | The prevalence of *S. mansoni* gradually dropped from 9.6% to 4.1. Nonetheless, a downward trend in *S. mansoni* was noted prior to the MDA's launch and persisted thereafter. The positivity rate was considerably higher in the 5–14 age group and in males. Significant seasonal variation was observed in *S.mansoni* infection in school-aged children. |
| 9 | [20] | Senegal | 777 children, Infected:226 Uninfected:551 | SAC 5–11 years | MDA 40 mg/kg WASH | Prospective cohort | *S. haematobium* | Kato Katz | School-based | High rates of infection reduction (between 96.7 and 99.7%) in the two research locations. The village that used the canal had a noticeably higher rate of re-infection. Praziquantel reduced the prevalence and intensity of urogenital schistosomiasis when administered periodically. |

*(Continued)*

**Table 1.** (Continued)

| SN | First author | Country | Sample size | Age group (Years) | Control strategy | Study Design | *Schistosoma* species | Diagnosis technique | Intervention (community or school-based or combined) | Summary of main findings Principal findings |
|---|---|---|---|---|---|---|---|---|---|---|
| 10 | [21] | Cote' d'Ivoire | 6400 individuals | SAC 5–14 years | MDA 40 mg/kg Snail control | Cluster randomized trial | *S. haematobium* | Reagent strip Urine filtration | School-based | Baseline prevalence was 12.5% and it decreased to 4.25% at year 4. Heavy intensity infections decreased from baseline to final survey among all age groups in all arms. At human water contact sites treated with 10g/L niclosamide, snails were dead the next day. *B. truncates* was the predominant species. Snail control did not significantly improve the effects of MDA |
| 11 | [32] | Cote d'Ivoire | 2455 individuals Infected: 618 Uninfected: 1837 | SAC 9–12 years | MDA 40 mg/kg | Cluster randomized trial | *S. mansoni* | Kato Katz | School-based | The prevalence decreased from 20.9% at baseline to 15.4% after 4 years MDA. Annual and biennial treatment plans yielded the same results and from the year to year 4 treatment coverage increased from 56% to 74%. |
| 12 | [23] | Tanzania | 1451 individuals 708 intervention 743 non-intervention | SAC Adolescents 9–16 years | Education, Behaviour change | Cluster randomized trial | *S. haematobium* | Urine filtration | School-based | Improvements in knowledge about S. haematobium transmission and perceptions of risk, as well as improved attitudes towards the prevention and treatment of the disease, with an increased uptake of swallowing anthelminthic tablets during MDA campaigns—which suggests that a group of children had been resistant to or did not swallow the drugs in previous MDA efforts prior to the intervention—and self-reported changes in behaviour are all significantly associated with exposure to behavioural interventions against urogenital schistosomiasis, guided by the social-ecological framework and grounded in constructs of health education. |
| 13 | [33] | Kenya | 1374 individuals | SAC 7–8 years | MDA 40 mg/kg | Repeated cohort study | *S. mansoni* | Kato Katz Abdominal Ultrasonography | School-based Community-based | At year one the prevalence was 64% infected at baseline-45.5% after the 5 year intervention. Study shows that treating schoolchildren on a regular basis is linked to lower rates of S. mansoni infection, as well as lower mean infection intensities. Both treatment groups exhibited improvements in health-related quality-of-life scores. |

*(Continued)*

**Table 1.** (Continued)

| SN | First author | Country | Sample size | Age group (Years) | Control strategy | Study Design | *Schistosoma* species | Diagnosis technique | Intervention (community or school-based or combined) | Summary of main findings Principal findings |
|---|---|---|---|---|---|---|---|---|---|---|
| 14 | [24] | Madagascar | 286 individuals | SAC 5–14 years | Health Education MDA 40 mg/kg | Cross sectional study | *S. mansoni* | Kato Katz | School-based | Three stages of design: participation, engagement, and community consultation. Between 2017 and 2018, the percentage of pre-education answers that were correct increased from 53% to 72%, with older children being more likely to give accurate answers. After the Schistosomiasis Education Program (SEP), children from participating schools had better answers on schistosomiasis surveys, improved latrine usage, and 91% MDA attendance—more than twice as many as those from SEP-naïve schools. MDA attendance decreased from 74% (2016) to 42% (2017) in schools without SEPs. |
| 15 | [34] | Kenya | 1110 individuals | SAC Adolescents 6–17 years | MDA 40 mg/kg | Cross sectional study | *S. mansoni* | Kato Katz | School-based | Annual school-based MDA to reduce the baseline proportion of infected children to 14.0% in Year 4 from 44.7%. Heavy infections decreased from 6.% to 0.3%. Four rounds of annual PZQ MDA significantly reduced *S. mansoni* infection prevalence and virtually eliminated heavy infections as defined by WHO guidelines. |
| 16 | [22] | Tanzania | 9700 individuals | SAC 9–12 years | MDA 40 mg/kg Snail control Behaviour change | Cluster randomized trial | *S. haematobium* | Urine filtration | School-based | 11 round of MDA in 7 years with 16 month treatment gaps. The prevalence of *S. haematobium* in schoolchildren dropped from 6.6% in 2012 to 3.4% in 2020, while the percentage of children with microhaematuria also decreased from 9.5% to 5.2%. In 2020, the considerable rebound in the overall prevalence in 2019 it was 2.8% and it increased significantly in 2020 to 9.1% in boys aged 9–16 years and also infection intensity was caused by certain re-infections in hotspot areas. |
| 17 | [35] | Nigeria | 434 individuals 108 infected 326 uninfected | SAC, Adolescents 5–17 years | MDA 40 mg/kg | Longitudinal study | *S. haematobium* | Urine filtration | School-based | Baseline prevalence was 24.9% and it decreased significantly to 2.1% at 6 months post treatment but a year later the prevalence became 7.7%. This might have resulted from re-infections, fresh infections that occurred after the start of treatment, or the presence of juvenile S. haematobium infections in some of the kids. |
| 18 | [36] | Ethiopia | 4286 SAC individuals Sex matched | SAC 5–14 years | MDA 40 mg/kg | Cross sectional study | *S. haematobium* | Urine filtration | School-based | In the current study, praziquantel's overall treatment coverage against schistosomiasis was 75.5%. The percentage of SAC children who received praziquantel treatment was considerably higher for those who went to school (84.1%) as opposed to those who didn't (14.4%). Boys where 27% more likely to swallow the drug compared with girls, |

(*Continued*)

**Table 1.** (Continued)

| SN | First author | Country | Sample size | Age group (Years) | Control strategy | Study Design | *Schistosoma* species | Diagnosis technique | Intervention (community or school-based or combined) | Summary of main findings Principal findings |
|---|---|---|---|---|---|---|---|---|---|---|
| 19 | [15] | Zimbabwe | 535 SAC | PSAC | MDA 40 mg/kg | Prospective cohort study | *S. haematobium* | Urine filtration | community-based | Prevalence significantly reduced from 24.9% at baseline to 2.8% at 12 months. The infections detected at 12 months were re-infections that increased as the samples were collected during hot dry seasons that encourage cercarial shedding and human water contact increased during that season. The risk of being infected with schistosomes in pre-school-aged children was noted to increase with increasing age. |
| 20 | [16] | Kenya | 400 PSC | PSAC | MDA 40 mg/kg | Longitudinal pre- and post-test study | *S. haematobium* | Urine filtration Kato Katz | School-based | *S. haematobium* infections were prevalent overall both before and after treatment, at 20.0% and 2.8%, respectively. According to the study's findings, pre-schoolers in Kwale County had a higher prevalence of S. haematobium (20%) than school-aged children in the same county (24.5%). The findings indicate that following treatment, 10 of the 80 children who had tested positive for *S. haematobium* had a light infection (1–49 eggs/10 ml urine). Before receiving treatment, these kids had infections with a high intensity (50 eggs per 10 millilitres of urine). |
| 21 | [37] | Sudan | 1951 SAC | SAC 6–14 years | MDA 40 mg/kg | Longitudinal cohort study | *S. haematobium S. mansoni* | Urine filtration | School-based | At baseline the prevalence was 13.8% and at 6 months the prevalence decreased to 4.2%. Re-infection rate infection was 9.8% at 6 months post treatment in high infection areas. Compared to low-infection areas, the rate of reinfection was significantly higher in high-infection areas (p = 0.02). |
| 22 | [18] | Tanzania | 20389 individuals | PSAC SAC Adolescents 3–17 years | MDA 40 mg/kg | Cross sectional study | *S. haematobium* | Urine filtration Dipstick | School-based | Overall, mean infection prevalence was 7.4% after 15 years of MDA at baseline it was 20.7%. Light and heavy infections were detected in 82.3% and 17.7% of the positive children respectively, Prevalence of schistosomiasis decreased post-treatment with schistosomiasis |
| 23 | [38] | Nigeria | 1,267 individuals | SAC Adolescents 10–17 years | MDA 40 mg/kg | Cross sectional study | *S, haematobium S. mansoni* | Urine filtration Kato Katz | Community-based | The average overall prevalence of S. haematobium in the ten states under study was 10.4%. The high intensity seen in some of the states indicates that there may still be a high rate of transmission and that some of the reported infected children either missed treatment or became re-infected after receiving treatment. |
| 24 | [39] | Senegal | 777 individuals | SAC 5–11 years | MDA 40 mg/kg | Prospective cohort study | *S. haematobium* | Urine filtration | Community-based | Baseline schistosomiasis prevalence was 59.7% it significantly decreased to 7.1%. Praziquantel has an effect on urogenital schistosomiasis prevalence and severity. However, in the Senegal river basin, *S. haematobium* remains a problem due to a resurgence in infection at 32.3%. |

(*Continued*)

**Table 1.** (Continued)

| SN | First author | Country | Sample size | Age group (Years) | Control strategy | Study Design | *Schistosoma* species | Diagnosis technique | Intervention (community or school-based or combined) | Summary of main findings Principal findings |
|---|---|---|---|---|---|---|---|---|---|---|
| 25 | [40] | Zimbabwe | 14000 individuals | SAC 6–15 years | MDA 40 mg/kg | Longitudinal study | *S, haematobium S. mansoni* | Urine filtration Kato Katz | School-based | The sentinel sites had a 31.7% S haematobium prevalence prior to the MDA. The most common schistosome species in the nation was S. haematobium. After six yearly MDA rounds, the prevalence of S. haematobium dropped to zero percent. |
| 26 | [41] | Ethiopia | 499 children from two schools Infected: 234 Uninfected: 265 | SAC, Adolescents 5–18 years | MDA 40 mg/kg | Cross sectional study | *S. mansoni* | Kato Katz | School-based | Baseline prevalence of schistosomiasis was 52.1%, after MDA the cure rate of 91.7% and reduced the egg rate by 86.8% was obtained. The efficacy of praziquantel at 40 mg/kg is sufficient to permit continued use in treating *S. mansoni* infected schoolchildren. |
| 27 | [42] | Tanzania | 1716 student cohort | SAC | MDA 40 mg/kg | Longitudinal cohort study | *S, haematobium* | Urine filtration | School-based | Due to high rates of re-infection, the MDA program only partially controlled parasite infections.. MDA was not as effective as the overall reduction in infection was statistically insignificant at 1.8% p = 0.1751. |
| 28 | [17] | Angola | 67 children Infected:47 Uninfected: 20 | PSAC SAC (2 - <15yrs) | MDA 40 mg/kg | Interventive prospective study | *S. haematobium* | Urine filtration | Community-based | At baseline schistosomiasis prevalence was 75.81%, at one month post treatment it decreased to 53.7% but at 6 months post treatment the prevalence rose to 76.1% the reinfection was 85.7%. The use of MDA to treat schistosomiasis is not very effective, reinfections happen quickly, and using anthelmintic therapy alone is not a long-term solution. |
| 29 | [43] | Togo | 17,100 children at 1129 schools | SAC Adolescents 5–17 years | *MDA* 40 mg/kg | Cross sectional study | *S. haematobium S. mansoni* | Urine filtration Kato Katz | Community-based | MDA coverage was 94% and baseline prevalence of schistosomiasis was 23.5% to 5.0% after 5 years of MDA. Both the frequency and severity of schistosomiasis infection were markedly decreased, and discontinuing MDA in regions where the infection is highly prevalent could cause a marked recurrence of the illness. |
| 30 | [44] | Niger | 380 children Infected: 55 | SAC 5–14 years | *WASH* | Cross sectional study | *S. haematobium* | Urine filtration | School-based Community-based | The prevalence of schistosomiasis was 14.5% compared to 51% they got a borehole but they were using one big stream before provision of portable water. |

and SAC [17]. Only one (n = 1) study focused on all three age groups [18]. The review also analysed the type of *Schistosoma* species investigated in the control-program-related studies in a test-to-treat approach and most of the studies were focusing on *S. haematobium* (n = 12) followed by *S. mansoni* (n = 10) and both species (n = 8) respectively. Schistosomiasis control strategies were predominantly MDA only in 24 studies and one study each on WASH only [19]; MDA and WASH [20]; MDA and snail control [21]; MDA, behaviour change and snail control [22]; education and behaviour change [23]; and MDA and education [24]. In terms of

the diagnostics techniques applied, the majority of studies utilised Kato Kats (n = 10) followed by urine filtration (n = 9); a combination of Kato Katz and urine filtration (n = 6); reagent strips and urine filtration (n = 1) and one study each in Kato Katz and PCR; Kato Katz and ultrasonography; and urine filtration, ELISA and Kato Katz. Articles were also assessed and analysed on the type of intervention (community or school-based or combined and most of the studies were predominantly school-based (n = 19) while the rest were either community-based (n = 8) or combined (n = 3).

## Geographic distribution of schistosomiasis control programs in sub-Saharan Africa

Although WHO reports showing immense progress in control programs in Africa, our current study showed that a low turn-out number of 11 from 55 sub-Saharan African countries were making concerted efforts in reporting evidence-based studies of schistosomiasis control programmes and most of the research is done in Kenya and Tanzania. However, the review includes studies from, Eastern(Kenya, Tanzania and Ethiopia), Western (Nigeria, Senegal, Togo, Cote' d'Ivoire and Niger), Southern (Zimbabwe and Madagascar) and Northern Africa (Sudan). Minimum schistosomiasis control activities are occurring in the Central and Southern regions of sub-Saharan Africa. Table 2 shows that MDA is the most implemented control strategy across sub-Saharan Africa, with WASH programs occurring in Northern, Eastern and Western Africa. Snail control and Health education control programs have been recorded in the Eastern and Western regions of sub-Saharan Africa.

Overall analysis included articles from the year 2016–2022 as shown in Fig 2. 2018, 2020 and 2022 had six studies each followed by 2021 with five studies. Both 2019 and 2017 had three studies included in the review with 2016 only having one.

**Study setting and design.** There were five longitudinal intervention studies [25,35,37,40,42], seven cluster randomized trials [23,24,27–29,32,35], one case control study [6], seven cross sectional studies [18,29,31,36,38,41,43,44], three prospective studies [15,19,33,39], two repeated cross sectional studies [24,34], longitudinal pre- and post- test design [16], one interventive prospective study [17] as shown in Fig 3. Tanzania and Kenya had the most intervention studies, they each had five intervention studies included in this review whilst Zimbabwe and Ethiopia have four intervention studies, Cote' d Ivoire has three intervention studies, Senegal and Nigeria had two intervention studies and Madagascar, Sudan, Niger, Angola and Togo each had one.

**Table 2. Schistosomiasis control strategies.** Different schistosomiasis control strategies in this review as per geographical regions.

| Schistosomiasis control strategy | Country and geographical region | References |
|---|---|---|
| Mass drug administration | Eastern Africa<br>Western Africa<br>Northern Africa<br>Central Africa<br>Northern Africa | [15–19,21–23,25–43,] |
| Water Sanitation and Hygiene (WASH) | Eastern Africa<br>Western Africa<br>Western Africa | [19,39,44] |
| Education and behaviour change | Eastern Africa | [22–24] |
| Snail control | Western Africa<br>Eastern Africa | [21,22] |

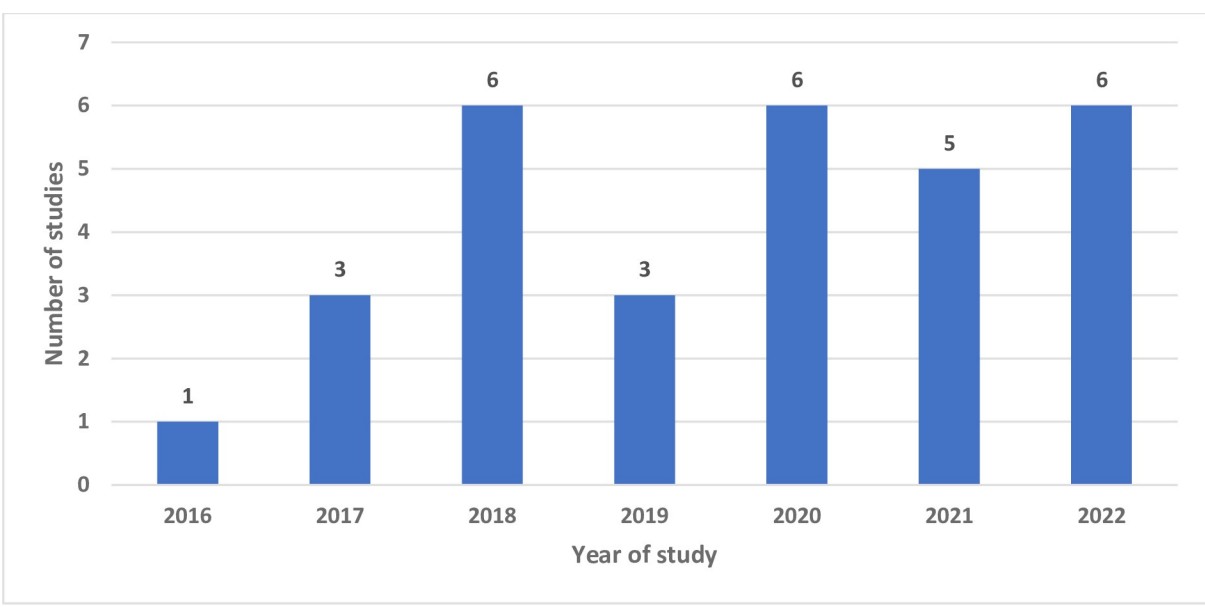

**Fig 2. An illustration of number of articles and publication years.**

**Study participants.**   The study was directed at the paediatric population in sub-Saharan Africa that has been subjected to schistosomiasis control programmes in the past 10 years. The current study participants' age groups ranged from 2–18 years with only four studies from Angola, Kenya, Tanzania and Zimbabwe including PSAC younger than 5 years old [6,15,16,18]. The rest of the studies had participants between the ages of 5–18 years. A total collective of 199 037 participants were included in the current review.

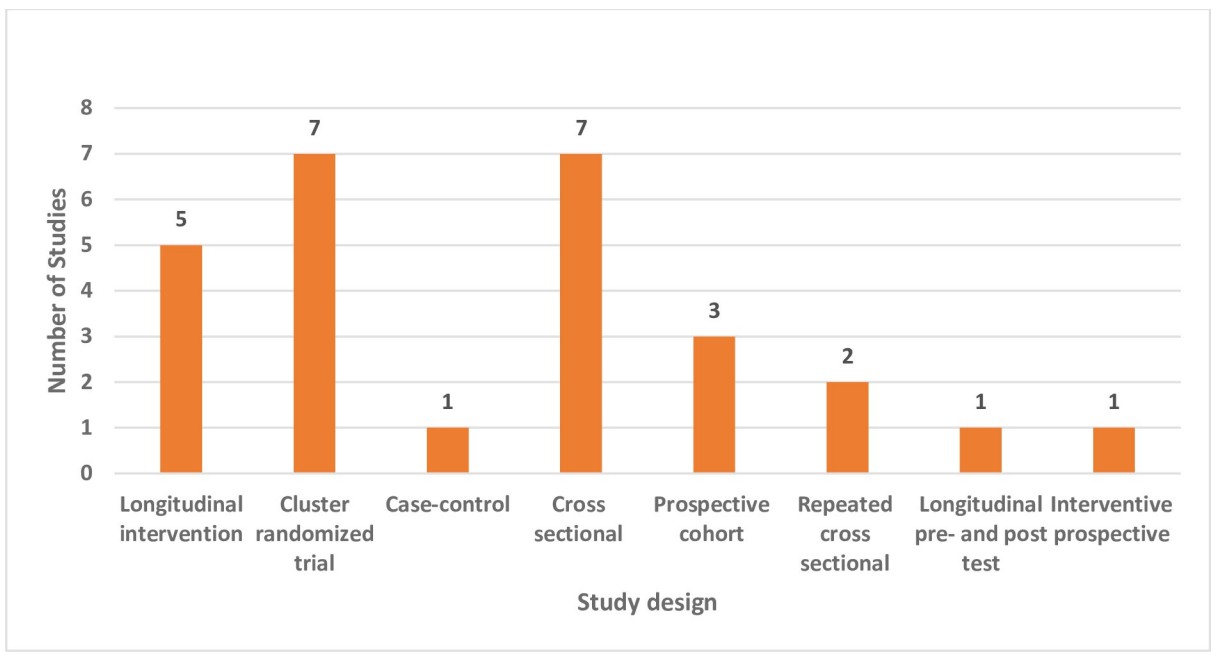

**Fig 3. Study designs.** An illustration of different types of study designs of articles used in the review.

**Diagnostic techniques.** The current study showed that the gold standard techniques Kato Katz and urine filtration were mainly used for the diagnosis of *S. mansoni* and *S. haematobium* infection respectively [6,26,38–40,44] and also used in the pre- and post-treatment test methods [16] as shown in Fig 4. One study, [29] successfully detected schistosome-specific DNA in *S. mansoni* and *S. haematobium* from filtered urine using PCR with high sensitivity and specificity with no cross-reactivity with other related parasites from a single source of the urine thus simplifying the collection and performance of tests.

**Schistosomiasis re-infections after MDA.** The current showed that after treatment with MDA there were statistically significant reinfections occurring in the treated paediatric population enrolled in the study [15,17,22,35,37,39,42]. Adewale et al 2018 also showed that even though there was a significant decrease in schistosomiasis infections in the study participants, post MDA programme there were significant re-infections [17]. Seven months after treatment, re-infection was observed. At this point, it was thought that learners had contracted *S. haematobium* again if they were releasing parasite eggs. Samples of urine were taken and analysed as previously mentioned (eggs/10 ml urine). Only children who tested positive at baseline, received treatment, and tested negative at the treatment assessment time point were monitored for reinfection. If, following the initial negative assessment, at least one egg was discovered, each person was deemed to be re-infected. After a year, the prevalence dropped dramatically from 24.9% at baseline to 2.8% [35]. When the samples were taken during hot, dry seasons that promote cercarial shedding and increase human water contact, the infections found at 12 months were re-infections that multiplied during that season.

It has been observed that as children get older, their vulnerability to contracting schistosomes increases. In the two study sites, high infection reduction rates (between 96.7 and 99.7%) were attained [20]. The village that used the canal had a noticeably higher rate of re-infection. When taken on a regular basis, praziquantel decreased the incidence and severity of urogenital schistosomiasis. The prevalence of schistosomiasis was 75.81% at baseline, dropped

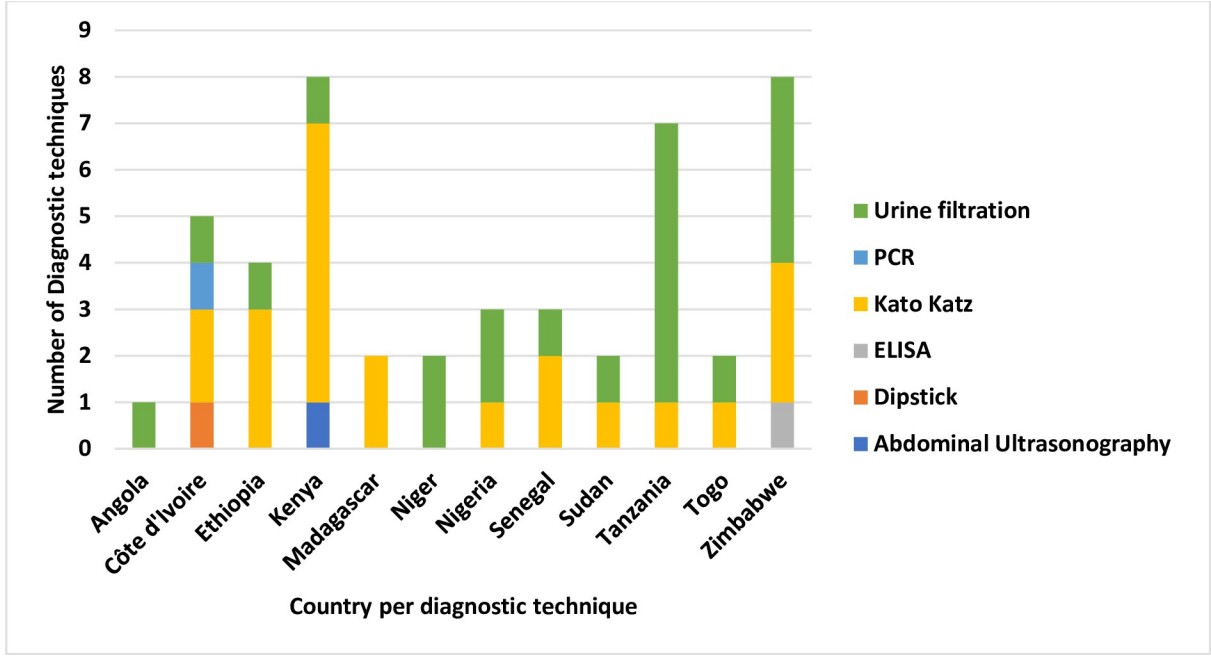

**Fig 4. Diagnostic techniques.** An illustration of diagnostic techniques used to detect schistosomiasis in each country involved in the review displayed as the number of times conducted in studies. Different colours are a representation of each diagnostic technique.

to 53.7% after one month of treatment, and increased to 76.1% after six months, with an 85.7% reinfection rate [17]. The use of MDA to treat schistosomiasis is not very effective, reinfections happen quickly, and using anthelmintic therapy alone is not a long-term solution [42].

## Effects of schistosomiasis control strategies

A total of 4 schistosomiasis control strategies were identified, as outlined below.

**Mass drug administration (MDA).** MDA was the main control strategy being employed by the sub-Saharan African countries either as preventive chemotherapy or in a test-to-treat intervention method. *S. mansoni* and *S. haematobium* are the two major strains that were tested and treated in all the included studies in this review, with *S. mansoni* causing more schistosomal infections than *S. haematobium*. The praziquantel dosage used in all the studies was 40mg/kg, and in most of the studies it caused a significant decrease in schistosomiasis prevalence.

In the overall analysis displayed in Table 3, schistosomiasis prevalence decreased significantly throughout most of the studies with participants infected with both *S. haematobium* [6,15,16,18,20,25,32,35,37,39,44] and *S. mansoni* [21,28,31,33,34,40]. Mduluza et al 2020 reached 0% after 6 rounds of MDA from a baseline prevalence of 31.7% with school-based intervention strategy at a 90% [40]. Lemos et al 2020 showed that MDA has low effectiveness to lowering prevalence of schistosomiasis with a prevalence difference of 0%, baseline

**Table 3. Duration of treatment programmes and percentage differences in pre-test and post-test during MDA programmes.**

| Study | | *S. haematobium* | | | *S. mansoni* | | |
|---|---|---|---|---|---|---|---|
| | Duration of treatment programme | Baseline % Prevalence | Post treatment % Prevalence | % difference | Baseline % prevalence | Post treatment % Prevalence | % difference |
| Trippler et al 2021 [21] | 7 years | 6.60 | 3.40 | -48% | | | |
| Ouattara et 2021 [31] | 4 years | 12.50 | 4.25 | -66% | | | |
| Jin et al 2021 [36] | 6 months | 13.80 | 4.20 | -70% | | | |
| Kimani et al 2018 [15] | 5 weeks | 20.00 | 2.80 | -86% | | | |
| Mazigo et al 2022 [17] | 15 years | 20.70 | 7.40 | -64% | | | |
| Kasambala et al 2022 [6] | 6 months | 22.00 | 3.60 | -84% | | | |
| Chisango et al 2019 [24] | 2 years | 23.10 | 0.47 | -98% | | | |
| Bronzan et al 2018 [42] | 5 years | 23.50 | 5.00 | -79% | | | |
| Adewale et al 2018 [34] | 12 months | 24.90 | 7.70 | -69% | | | |
| Mutsaka-Makuvaza et al 2018 [30] | 12 months | 24.90 | 1.80 | -93% | | | |
| Mduluza et al 2020 [39] | 6 years | 31.70 | - | -100% | | | |
| Kim et al 2020 [41] | 3 years | 43.90 | 4.80 | -89% | | | |
| Ekanem et al 2017 [43] | 1 year | 51.10 | 14.50 | -72% | | | |
| Senghor 2016 [19] | 3 years | 57.70 | 4.20 | -93% | | | |
| Senghor er al 2022 [38] | 9 months | 59.70 | 7.10 | -88% | | | |
| Lemos et al 2020 [16] | 6 months | 75.80 | 76.10 | 0% | | | |
| Mduluza et al 2020 [39] | 6 years | | | | 4.60 | - | -100% |
| Karanja et al 2017 [25] | 5 years | | | | 17.68 | 8.71 | -51% |
| Secor et al 2020 [26] | 5 years | | | | 33.00 | 17.43 | -47% |
| Olsen et al 2018 [27] | 5 years | | | | 54.60 | 45.20 | -17% |
| Zeleke et al 2020 [29] | 6 years | | | | 9.60 | 4.10 | -57% |
| Ouattara et al 2022 [20] | 4 years | | | | 20.90 | 15.30 | -27% |
| Shen et al 2019 [32] | 5 years | | | | 64.00 | 45.50 | -29% |
| Abudho et al 2018 [33] | 4 years | | | | 44.70 | 14.00 | -69% |

prevalence was 75.81%, one month post praziquantel administration the prevalence went down to 53.7% and after 6 months the prevalence increased to 76.1% [17]. Three studies occurred for a time below a year, four studies had one year MDA programme and the remaining MDA control programmes took more than a year. Only one MDA control program from Tanzania took 15 years to be completed and although there was decrease in schistosomiasis prevalence, at the end of the MDA programme, there was still 7% prevalence [18].

**Water, sanitation, and Hygiene (WASH).** The study showed that WASH control programmes were either integrated with MDA or focused only on WASH. In these two studies, Senghor et al [39] reported a high prevalence of schistosomiasis and the intensity of infection in children residing in locations using irrigation canals and rivers despite repeated rounds of chemotherapy and their proximity to these unsafe water facilities increased the incidence of infection. The later study further recommends that chemotherapy must be administered once every 2 years in SAC as stipulated by WHO as long as WASH status is not improved in such areas. Furthermore, Grimes and colleagues *2016] demonstrated that improving WASH status such as handwashing following defecation and increasing the number of latrines and primary water sources reduce transmission of schistosomiasis and each different form of WASH appears to have different effects on infection with various schistosome parasites [19].

**Education and behaviour change.** Education and behaviour change were found to be integrated in other main schistosomiasis control programmes namely MDA and snail control programs [24]. In a study where the education for behaviour change model was combined with MDA, Person et al [23] reported a significant change in children that had received treatment and transmission knowledge on schistosomiasis resulting in self-reported changes in risk behaviours such as washing or bathing in river streams. Additionally, there was a significant increase in swallowing praziquantel tablets during MDA campaigns hence reducing the prevalence of schistosomiasis. Spencer and colleagues [24] designed a low-cost novel schistosomiasis education that included cartoon books, games, songs, puzzles, and blackboard lessons resulting in an improvement in attendance from 64% to 91%, and latrine use from 89% to 96%. Sensitization, community consultation, engagement and participation was used in ensuring schistosomiasis education programmes in schools were fully undertaken and to encourage participation. Comparing schools that received schistosomiasis education, to those that didn't, it was observed that the education programmes improved latrine use, MDA uptake and understanding schistosomiasis transmission [24].

**Snail control.** Ouattara and colleagues [21] utilised niclosamide, 10g/L on a malacological survey against *Bulinus* species (*B. truncatus and B. globosus*) on human-water contact sites three times a year. and found that adding annual MDA to the layer of intermediate host snail control did not have a big significance in reducing the prevalence and intensity of schistosome infection. The main factors that hinder the success of snail control programs are the complexity of the strategy which includes water surface size to be treated with niclosamide, difficulties in identification of all human-water contact sites and sporadic application of niclosamide that does not stop the repopulation of treated areas by intermediate host snails.

**Intervention strategies and coverage.** School-based intervention strategy was predominantly applied in the selected studies which explains why most of the control programs were largely focused on SAC and adolescents. Schools are the main target sites during MDA due to the high numbers of children concentrated in a single place facilitating ease of drug dispersal and cost-effectiveness as shown in Fig 5 [27]. The remaining studies reported the benefits of integrating both community and school-based interventions in schistosomiasis control strategies. Secor et al. [27] compared school-based and community-based strategies during MDA and data support the use of biennial school treatment coupled with attention to infections in

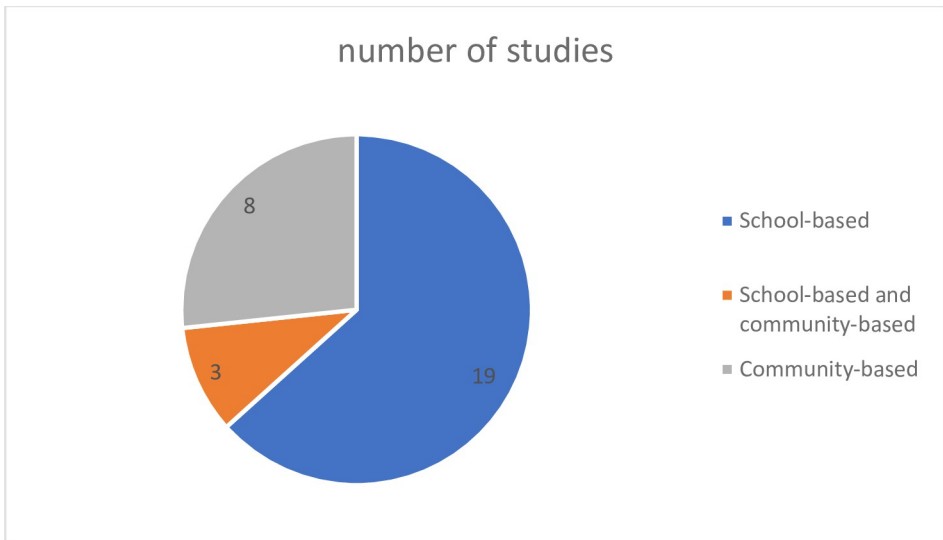

**Fig 5. Types of intervention strategy.** Illustration of the intervention strategy as school-based, community-based or both.

younger children and occasional treatment of adults in the community to yield a positive effect in the reduction of schistosome infection.

School-based treatment had higher coverage than community-based treatment even after household-to-household treatment. School-based treatment had >89% coverage whilst community-based treatment had <80% coverage [27]. MDA coverage above 75% was significantly linked to a decrease in prevalence if there was consistency during the entire control programmes. With lower coverage the prevalence and intensity of schistosomiasis infection would increase and stopping MDA in areas with high prevalence resulted in a significant rebound of infection [17].

## Discussion

This systematic review provided a summary of the effects of paediatric schistosomiasis control programs in sub-Saharan Africa. Various outcomes were noted after reviewing 30 articles on paediatric schistosomiasis, MDA programmes by preventive chemotherapy, and additional interventions including, WASH, snail control and health education. The review showed that despite preventive chemotherapy lowering schistosomiasis prevalence, chances of re-infection remained high in endemic areas. Similar findings were found in studies conducted in Sudan with an overall reinfection rate of 9.8% six months post MDA treatment [37], Nigeria with 7.7% reinfection rate at 12 months post treatment [35], and northwest Ethiopia, with a reinfection rate of 13.9% after six months post treatment [45]. According to WHO guidelines, the frequency of treatment and coverage levels are determined by the degree of infection in a specific geographic area. In high prevalence settings, or places where the baseline prevalence is 50% or higher, the current recommendations for SAC treatment are once a year. Treatment should be administered once every two years in settings with moderate prevalence (10% to 50% baseline prevalence) and once every three years in settings with low prevalence (<10% baseline prevalence). The studies done in Senegal, Tanzania and Angola had a baseline prevalence of more than 50% and the MDA control programmes ran for nine months, five years and six months respectively and the prevalence reduction was significant in the Senegal studies [17,28,39]. Trippler et al 2021 showed that 11 rounds of MDA during a seven year treatment programme

in Tanzania, with 16 month treatment gaps also did not reach a zero prevalence post treatment due to reinfections [21]. All the other studies had a moderate baseline prevalence (between 10–50%) and the reduction in prevalence was statistically significant after the different time periods.

The various schistosomiasis MDA programmes can be made more successful and sustainable if they are combined with continued health education and behaviour change programs to engage the target population and address specific features of the infection cycles [46]. Introducing health education into routine programmes can be one way of encouraging MDA intake, building knowledge to encourage preventive practices and attitudes in the communities using specific roadmaps that comprise of identification of cross-cutting approaches that emphasized awareness-building in the population and community in combination with other schistosome control strategies [47]. Behaviour change due to education programmes can also strengthen control by interrupting transmission through modifying exposure behaviour via water contact or transmission practices via open urination or defecation and through fostering treatment-seeking or acceptance [8]. However such school-based interventions leave out PSAC that do not attend school and are prone to infections at home and adults thus limiting the success of control and elimination programmes [48]. Operational difficulties, including obtaining parasitology samples for diagnosis, failure to detect light infections and inadequate knowledge about risk factors in PSAC, have been found to have biased interventions towards a school-based approach [49].

The danger of persistent re-infections in children who are repeatedly infected with the Schistosoma parasite is the development of anaemia, stunting and learning difficulties [2]. In addition, after years of infection, the parasite can damage various organs in the body, including the liver, intestine, lungs, and bladder. In rare cases, the eggs of the parasite can be found in the brain or spinal cord, which can cause seizures, paralysis, or inflammation of the spinal cord [50]. In schistosomiasis endemic areas there is persistent reinfection rates due to frequent water contact behaviours among school-aged children and environmental risk factors explain the variations in patterns of infection seasonally among school-aged children [47].Studies conducted in Nigeria, Zimbabwe, Sudan, Senegal and Tanzania confirmed reinfections therefore the importance of complementary interventions and addition of snail control and education on schistosomiasis prevention could help provide a more significant decrease in the infection prevalence and intensity [15,35,37,39,47]. Some indirect interventions such as making sure children go to school will reduce reinfections in highly endemic areas [51].

Although making up only 13% of the global population, up to 90% of all schistosomiasis cases occur in Sub-Saharan Africa. In order to totally control and eradicate the schistosome infections, concentrated efforts must be made, primarily in this burdened continent. The review indicates that including vector control in schistosomiasis control programmes has great potentials to disrupt schistosomiasis transmission especially in areas where there is high prevalence and noncompliance to MDA [52]. Niclosamide is a vector control chemical that has been used as molluscides in the studies included in the review as part of combined intervention strategies in study [20,22]. The results shows a better chance of controlling schistosomiasis and definitely reduced the chances of reinfections. Therefore despite the successful use of niclosamide in killing snails, their populations usually rebound quickly and if they come in contact with Schistosoma infested water bodies, the transmission 'hot spots' will be restored [53]. Copper sulphate and sodium pentachlorophenol were in use in the past while currently, WHO recommends niclosamide due to its low toxicity for humans and its ability to kill snails, their eggs and cercariae at very low concentrations [54]. For an effective snail control program, it is of paramount importance to know the local snail genera, their preferred habitats and the

proper application of molluscicide to achieve the desired effects [53]. Effectiveness of snail control is momentary as snails develop resistance to chemicals used in molluscidation.

The association between water contact and schistosomiasis infection is a major factor that has influenced the impact of schistosomiasis prevention in sub-Saharan Africa and should be strongly considered in prevention strategies [8]. Some communities in sub-Saharan regions for example, have considered water bodies to be sacred as they perform various, ceremonies and religious activities. In this instance, water contact is frequent regardless of its safety or hygiene. Similarly, water bodies in some areas such as rivers and lakes are a source of economic activity such as fishing and agriculture, and they are used for recreational activities such as swimming. In such cases, people will be more likely to encounter schistosomiasis infested water bodies, even if they are aware of the risks of contracting the disease. Therefore, it is important when designing prevention interventions, to understand the local context, knowledge, attitudes, and practices towards water bodies to reduce water contact and prevent transmission of schistosomiasis. Tailor made interventions will be more effective in reducing water contact and ultimately, prevent transmission of schistosomiasis as shown in studies implementing educational programs in Madagascar and Kenya [8,24].

Additionally, the review details the different diagnostic methods used in the test to treat interventions. The gold standard diagnostic tools which use microscopy to quantitatively analyse Schistosoma ova include Kato Katz for the detection of *S. mansoni* and urine filtration for the detection of *S. haematobium* have not been sensitive enough, especially in low-endemic settings. This has led to an underestimation of the true positives when determining the prevalence of schistosomiasis pre and post intervention. Schistosomiasis control interventions require a more targeted diagnostic tool to precisely estimate the prevalence of schistosomiasis just as illustrated in the study done by Amoah [55]. Sensitive diagnostic tests and targeted preventive chemotherapy is required to disrupt the transmission of schistosomiasis and eventually eliminate the disease [3]. Finally, in the paediatric population, proper water sanitation and hygiene activities should be made available, educating the teachers and caregivers to impart and provide knowledge on schistosomiasis, and associations between water contact and infections should be expressed to control and eliminate schistosomiasis. The study showed that, despite its high cost, PCR is a useful tool for identifying low-intensity infections, which will improve surveillance efficiency and evaluate the effect of MDA control initiatives against schistosomiasis. For tracking the effectiveness of treatment, a conclusive detection method is also an essential component of the regimen, both in the clinic and during MDA. The gold standard diagnostic test is microscopic examination of excreta in urine or stool; however, this method has low sensitivity, particularly when detecting light infections, and requires adult worms to be producing eggs [56]. However, the main drawbacks of serological and polymerase chain reaction (PCR) testing are their high costs and lengthy turnaround times. On the other hand, these methods can identify less severe infections with excellent specificity and accuracy.

It will take a coordinated effort and plan to ensure that the most affected countries have strong clinical, public health, laboratory, pharmaceutical, and research capacities to ensure that the last mile towards elimination is not only reached but sustained. This includes the elimination of NTDs, including schistosomiasis. In order to develop and strengthen schistosomiasis programs in all impacted countries, this entails increasing strategic investments. The majority of schistosomiasis infections occur in low-income nations with unique political and cultural characteristics, inadequate health care, and compromised health systems made worse by the COVID-19 pandemic. In order to control schistosomiasis in all endemic areas of the affected countries, the agenda needs to be led [57,58]. Although there are high chances of re-infection in sub-Saharan Africa, preventive chemotherapy is an effective way to eliminate schistosomiasis if the WHO guidelines on praziquantel administration in endemic areas are

followed. However, preventive chemotherapy must be combined with other schistosomiasis control programmes such as improved sanitation, hygiene, snail control, behaviour change and positive attitudes and practices that can collectively achieve the end of transmission.

## Recommendations

To achieve schistosomiasis elimination and reach end of transmission, precision mapping, expanding geographical coverage and improving methods of chemotherapy delivery to all age groups is necessary [27]. To guarantee that younger patients take the medication, access to the upcoming paediatric praziquantel formulation must be increased. To promote medication uptake, more convenient ways to give praziquantel to the PSAC should be created. Additionally, this comprehensive strategy should occur alongside a more formal inclusion of these groups in large-scale monitoring and evaluation activities. Current omission of these age groups especially PSAC from most treatment plans and monitoring has been reported to exacerbate health inequities which lead to long-term consequences for sustainable schistosomiasis control [24,53]. Treatment could be administered more frequently than annually and an increased treatment regime should be introduced to cover wider time periods and combat reinfection rates as much as possible [28]. Combining intervention strategies should be the a standard procedure when designing and implementing schistosomiasis control programmes. WASH strategies should become a main priority in the elimination programmes as they are just as important as MDA programmes. Paediatrics also require health education programmes to increase awareness of schistosomiasis and effects of infections in a way that they can understand. Prevalence of infection and associated risk factors must be established to achieve total coverage of the PSAC and SAC groups in MDA and schistosomiasis control activities [20]. Highly sensitive diagnostic tests should be available in the test to treat approach so as to treat the individuals who truly require treatment [53].

## Study strengths and limitations

The review provides a comprehensive assay of the effects of paediatric schistosomiasis control programmes in sub-Saharan Africa with emphasis on intervention strategies, diagnostic tests, and methods of MDA interventions. The systematic reviews followed a predefined protocol by PRISMA, which makes the results reproducible and transparent. The JBI critical appraisal tools were used to assess study quality and multiple reviewers screened the articles in a rigorous and transparent methodology to minimize the risk of publication bias while increasing the validity of the results. The limitations of the review include focusing on the paediatrics in sub-Saharan Africa only therefore, the findings may not be generalizable to other age groups or regions. Furthermore, the reviews' included studies varied in quality, which might have an impact on the review's overall findings. Despite taking a lot of time, publication bias may have also had an impact on the systematic review because researchers typically publish positive results more frequently than negative ones. There were fewer studies on snail control, WASH activities and health education and this may have provided limited evidence on the said topic. This review was also limited by heterogeneity, as the different intervention strategies included were different in terms of design, population, and outcomes, which made it difficult to compare and synthesize the results.

## Conclusion

This review provided an overview of the effects of paediatric schistosomiasis control programmes in sub-Saharan Africa to reduce and eliminate schistosomiasis among the paediatric population. Findings shown that preventive chemotherapy with praziquantel is an effective

intervention strategy that lowers infection intensity and prevalence of schistosomiasis through MDA programs. However, it is not remotely enough to eliminate schistosomiasis or sustain low-intensity infections on its own due to reinfections in treated children and low coverage of the MDA programs with the low funding that is unable to cover every endemic area in sub-Saharan Africa. Hence schistosomiasis prevention requires a variety of complementary control strategies such as snail control, WASH activities and behavioural change modifications to significantly reduce and ultimately end disease transmission. The findings can be used by policy makers to assist in the elimination of paediatric schistosomiasis in sub-Saharan Africa.

## Supporting information

**S1 Table. PRISMA 2020 abstract checklist.**
(PDF)

**S2 Table. PRISMA 2020 checklist.**
(PDF)

**S1 File. Articles assessed for eligibility.**
(PDF)

**S2 File. Excluded articles.**
(PDF)

**S3 File. Final studies included in this review.**
(PDF)

**S4 File. Search strategy.** This is the search strategy that was used in review's search process for articles relevant to the study.
(PDF)

## Author Contributions

**Conceptualization:** Maryline Vere, Wilma ten Ham-Baloyi, Paula Ezinne Melariri.

**Data curation:** Maryline Vere, Wilma ten Ham-Baloyi, Paula Ezinne Melariri.

**Formal analysis:** Maryline Vere, Wilma ten Ham-Baloyi, Paula Ezinne Melariri.

**Funding acquisition:** Wilma ten Ham-Baloyi, Paula Ezinne Melariri.

**Investigation:** Maryline Vere, Wilma ten Ham-Baloyi, Paula Ezinne Melariri.

**Methodology:** Maryline Vere, Wilma ten Ham-Baloyi, Paula Ezinne Melariri.

**Project administration:** Maryline Vere, Wilma ten Ham-Baloyi, Paula Ezinne Melariri.

**Resources:** Wilma ten Ham-Baloyi, Paula Ezinne Melariri.

**Software:** Maryline Vere.

**Supervision:** Wilma ten Ham-Baloyi, Paula Ezinne Melariri.

**Validation:** Wilma ten Ham-Baloyi, Paula Ezinne Melariri.

**Visualization:** Maryline Vere, Wilma ten Ham-Baloyi, Paula Ezinne Melariri.

**Writing – original draft:** Maryline Vere.

**Writing – review & editing:** Maryline Vere.

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
