## [Decision Letter · Decision Letter 0]

3 Oct 2023

PONE-D-23-22834Impact of paediatric schistosomiasis control programmes in sub-Saharan Africa: A systematic reviewPLOS ONE

Dear Dr. Vere,

Thank you for submitting your manuscript to PLOS ONE. After careful consideration, we feel that it has merit but does not fully meet PLOS ONE’s publication criteria as it currently stands. Therefore, we invite you to submit a revised version of the manuscript that addresses the points raised during the review process.

We look forward to receiving your revised manuscript.

Kind regards,

Raquel Inocencio da Luz, Phd

Academic Editor

PLOS ONE

Journal Requirements:

- https://www.ncbi.nlm.nih.gov/pmc/articles/PMC4377019/

- https://www.frontiersin.org/articles/10.3389/fitd.2023.1116831/full

- https://bmcinfectdis.biomedcentral.com/articles/10.1186/s12879-022-07829-x

- https://journals.plos.org/plosone/article?id=10.1371%2Fjournal.pone.0189400

In your revision ensure you cite all your sources (including your own works), and quote or rephrase any duplicated text outside the methods section. Further consideration is dependent on these concerns being addressed.

3. We note that Figure 2 in your submission contain map images which may be copyrighted. All PLOS content is published under the Creative Commons Attribution License (CC BY 4.0), which means that the manuscript, images, and Supporting Information files will be freely available online, and any third party is permitted to access, download, copy, distribute, and use these materials in any way, even commercially, with proper attribution. For these reasons, we cannot publish previously copyrighted maps or satellite images created using proprietary data, such as Google software (Google Maps, Street View, and Earth). For more information, see our copyright guidelines: http://journals.plos.org/plosone/s/licenses-and-copyright.

1.) You may seek permission from the original copyright holder of Figure 2 to publish the content specifically under the CC BY 4.0 license.  

2.) If you are unable to obtain permission from the original copyright holder to publish these figures under the CC BY 4.0 license or if the copyright holder’s requirements are incompatible with the CC BY 4.0 license, please either i) remove the figure or ii) supply a replacement figure that complies with the CC BY 4.0 license. Please check copyright information on all replacement figures and update the figure caption with source information. If applicable, please specify in the figure caption text when a figure is similar but not identical to the original image and is therefore for illustrative purposes only.

Reviewers' comments:

Reviewer's Responses to Questions

**Comments to the Author**

1. Is the manuscript technically sound, and do the data support the conclusions?

Reviewer #1: Yes

Reviewer #2: No

Reviewer #3: Yes

2. Has the statistical analysis been performed appropriately and rigorously? 

Reviewer #1: Yes

Reviewer #2: N/A

Reviewer #3: N/A

3. Have the authors made all data underlying the findings in their manuscript fully available?

Reviewer #1: Yes

Reviewer #2: Yes

Reviewer #3: Yes

4. Is the manuscript presented in an intelligible fashion and written in standard English?

Reviewer #1: Yes

Reviewer #2: Yes

Reviewer #3: Yes

5. Review Comments to the Author

Reviewer #1: Dear Authors,

Your work is extremely important, especially now when schistosomiasis endemic countries are working towards controlling and eliminating the disease.

The review followed the standard guidelines and was comprehensive. I only have minor suggestions.

1. Lines 215-218 - Is it possible to state the specific countries in the mentioned regions?

2. Line 227-Give a brief sentence of what reference 47 is about.

Best Wishes.

Reviewer #2: Thanks for your draft, below you can find my observations to this work.

INTRODUCTION:

a. I'd urge authors to be careful with various strong statements observed in the text:

- Lines 50-53: "there is decent exposure in pre-school aged children (PSAC) (≤5 years), in spite of the age group being excluded from schistosomiasis studies". That is not correct. PSAC have often been studied. Unclear what the statement indicates.

- Lines 65-68: "With 75% coverage, three years of annual treatment of SAC with praziquantel could achieve elimination", that cannot be asserted conclusively as in the text.

b. Parts of the narrative are not clear:

- Lines 74-78: It alludes to a review of impacts of MDAs without explicitly mentioning the reference. The text remains unclear.

- Line 78-83: It is unclear why a review of snail control is expected to allude to the impact of friendly dosages of praziquantel for children.

c. Overall observation - Lack of stated objectives & contribution: In no part of the introduction there is clear statement indicating what is the specific objective of the review and what would be the academic contribution to ongoing debates of paediatric treatment for schistosomiasis control and elimination.

METHODS:

a. Methodology consistency: The review indicates that it only accepted peer-reviewed publications but at the same time the search included the review of grey literature. Were those references included in the end somehow? Why were those searches considered necessary?

b. Review process: The authors indicated that they made a screening of papers based on titles only and then proceeded to screen based on abstracts. That's highly problematic since titles do not provide sufficient evidence to discard publications. The entire selection process of this review would need to be revised as a result.

c. Overall observation - Lack of clarity about outcome measures: Generally speaking, the authors declare they would like to assess the impacts or effectiveness of paediatric interventions for SCH control / elimination. However, at no moment they specify what they mean by "impacts / effectiveness". There is no operationalisation of outcome measures and hence there is no clarify in any section of the paper of what they are trying to assess. This is profoundly problematic for the overeall structure of the review.

RESULTS:

a. PRISMA chart inconsistency: The PRISMA chart indicates that there were 198 studies selected for full asessment and that 158 were excluded. If that's the case, the review would end up discussing 40 publications, not 30 as stated in the text. The consistency of the charts needs to be reviewed and amended.

b. Graphs: There is no clarity as to why all graphs provide country-based comparisons / descriptions if the results section does not proeed to assess such differences. It is unclear what information they are trying to provide and how this is relevant to the overall study.

c. Overall observation - Results section fails to actually ellaborate on the results: The section includes substantive opinion and discussion. Actual references to the studies reviewed, beyond what is contained in the summary table, is minimal. This is not standard for a results section that demands authors to focus on expanding their findings from their review. The various references to opninion pieces, recommendations, or policy / technical guidelines should be moved to the discussion section. The entire results section should be revised in this regard.

DISCUSSION:

a. Overall concern: I am afraid that the discussion section is, as it stands, not fit for purpose. Since the results section has failed to provide any meaningful summary of key messages and since the overall assessment has not specific any specific outcomes measures to be revieweed, there are many uncertainties with regards to what is the core message that is being ellaborated and what pieces of evidence may support it. Some examples of problematic assertions guiding the discussion section can be found below:

- Lines 365-367: "The review showed that despite preventive chemotherapy lowering schistosomiasis prevalence, chances of re-infection remained high in endemic areas" This is not entirely accurate. Re-infection is mentioned in the table but not in the core results narrative. In addition, there's no clarity as to how the risk of reinfection was assessed in this review. Some references sometimes only mention re-infection without providing any specific indicators.

- Lines 379-382: "Preventive chemotherapy must be combined with other schistosomiasis control programmes such as improved sanitation, hygiene, snail control, behaviour change and positive attitudes and practices that can

collectively achieve the end of transmission." Generally, I agree with the sentiment. However, the evidence provided in the review neither supports such a strong assertion nor it is really referred to in the results section. Note, morever, that the Zanzibar studies that you consider support this statement in fact do not provide any conclusive evidence of the kind. This is either a misinterpretation of the results of such studies or a misuse of such references.

- Lines 389-392: "The review indicates that including vector control in schistosomiasis control programmes will definitely lead to end of transmission especially in areas where there is high prevalence and noncompliance to MDA" Same as previous paragraph, the body of evidence to back such a strong claim is not presented in the text.

b. One final coment is that, as mentioned in the introduction section, there is a lack of clarity of what the contribution of the review is. The authors should make an effort to assess how and to what extent their results differ or expand on existing overall recommendations for elimination.

Reviewer #3: Dear authors,

Certainly, the article is extremely important for the epidemiological context of schoolchildren in Africa, since schistosomiasis is still an important public health problem. However, some considerations should be made in the text to make it clearer.

Introduction

1) As the authors robustly described the infection in children, I suggest making it clearer in the introduction what the upper limit of the age range of schoolchildren.

2) Line 63-64 – authors could consider exploring what environmental improvements should be implemented as prevention for reinfection. For example, WASH interventions.

Materials and method

A few methodological points for the authors to consider:

3) It is appreciable and important that the authors took care to register the systematic review in the PROSPERO database. However, it would be very important to describe some key information in methodological writing for readers, such as the guiding question of the work and the temporality or years considered.

4) The authors mention that several terms were used in the research, however, in the main text it would be more useful to leave the details to the supplementary file outline and cite the supplementary file in the text. Describe the groups of terms used: for example, the following search terms was used: schistosomiasis OR agents AND control strategies. And write a sentence about which control strategies were used. Were no secondary terms used in the search to target countries of interest?

5) Inclusion criterion 3: please, consider explaining better what types of case study/series? Was a clinical case report article considered? Or was it considered a case report related to a specific community, neighborhood, or city? Since the authors talk about the impact of control programs, it would be of interest to also include information in the main text of the methodology, which study designs were considered.

6) In the description of data extraction, there was no behavioral information. Please consider describing that the information extracted considered behavioral data related to the disease.

7) Line 120-126: please, consider mentioning that the inclusion and exclusion criteria defined for the study were observed in all phases.

8) Line 135-136: evaluate and place this information in the “results” section.

9) Line 139-144: this information refers to the “selection process” section.

10) Line 149: was only intervention studies considered? Observational with comparison groups?

11) Line 153: were interventions at the household level not considered? Please, justify.

Results

12) Fig1.: please, consider describing the number of articles for each of the exclusion reasons (n=3257).

13) In the extracted data, there was no mention of extracting data on behavioral changes.

14) Consider describing the publication dates of the studies in one sentence and the follow-up date of the study control program; The review process and the temporal effects of a disease control program are crucial.

15) Please, consider better describing the scenario before and after implementing program control actions. The title of the article presupposes an assessment of “impact” on infection rates, which leads us to understand that there would be “before” and “after” evaluation in relation to the application of the program/strategy. Maybe use “effect” instead of “impact”.

16) Consider using the description of strategies (text after table 1) to describe with more focus what was used in the included studies. For example, the included study (23) used behavioral change strategies, but in the text of lines 283-302 there is a more general description.

17) There was no more detailed description of the methodological section in the “Critical Appraisal and Risk of Bias” section. Please, consider including in the text or supplementary material with indication in the text.

Discussion

18) Line362: the authors use the term “effects” and not “impact”. Please standardize and consider revising the term in the article title.

19) The authors make a very satisfactory and important discussion from the point of view of the findings. The article is important because it is the first to address the topic in the region and an integrated pediatric strategy.

20) Recommendations (Lines 428-438): please, consider including recommendations for all features described in Line 150.

6. PLOS authors have the option to publish the peer review history of their article (what does this mean?). If published, this will include your full peer review and any attached files.

Reviewer #1: No

Reviewer #2: No

Reviewer #3: No

---

## [Author Response · Author response to Decision Letter 0]

11 Dec 2023

Reviewer #1

Your work is extremely important, especially now when schistosomiasis endemic countries are working towards controlling and eliminating the disease.

The review followed the standard guidelines and was comprehensive. I only have minor suggestions. 

Thank you for your valuable and kind feedback – we appreciate it

1. Lines 215-218 - Is it possible to state the specific countries in the mentioned regions? Addressed 

Line 217-219

2. Line 227-Give a brief sentence of what reference 47 is about. Reference was removed as Figure 2 was replaced. 

Reviewer #2

Thanks for your draft, below you can find my observations to this work.

INTRODUCTION

a. I'd urge authors to be careful with various strong statements observed in the text:

- Lines 50-53: "there is decent exposure in pre-school aged children (PSAC) (≤5 years), in spite of the age group being excluded from schistosomiasis studies". That is not correct. PSAC have often been studied. Unclear what the statement indicates.

- Lines 65-68: "With 75% coverage, three years of annual treatment of SAC with praziquantel could achieve elimination", that cannot be asserted conclusively as in the text.

 Lines 50-53

Changed to:

Schistosomiasis control programmes in PSAC for the past years has been limited due to a lack of data on the safety and specificity of praziquantel dosage for this specific age group.

Lines 65-78

Removed elimination to:

Control of morbidity

b. Parts of the narrative are not clear:

- Lines 74-78: It alludes to a review of impacts of MDAs without explicitly mentioning the reference. The text remains unclear.

- Line 78-83: It is unclear why a review of snail control is expected to allude to the impact of friendly dosages of praziquantel for children.

 Lines 74-78 and Lines 78-83

The reason the review of snail control was indicated was to show that systematic reviews that have been done in the field of schistosomiasis control, have been lacking , their are research gaps in the study area thereby building up to the need for a systematic review to determine the effects of schistosomiasis control programmes in school aged children. 

c. Overall observation - Lack of stated objectives & contribution: In no part of the introduction there is clear statement indicating what is the specific objective of the review and what would be the academic contribution to ongoing debates of paediatric treatment for schistosomiasis control and elimination. Objective: to determine the effects of different schistosomiasis control programmes in the paediatric population in sub-Saharan Africa

The contribution was also added

METHODS:

a. Methodology consistency: The review indicates that it only accepted peer-reviewed publications but at the same time the search included the review of grey literature. Were those references included in the end somehow? Why were those searches considered necessary?

 None of the articles from the grey literature were actually included in the review as stated under ‘Search and selection results’. We have indicated that the manual search for grey literature and a citation search was to ensure a comprehensive search (see ‘Search strategy’)

b. Review process: The authors indicated that they made a screening of papers based on titles only and then proceeded to screen based on abstracts. That's highly problematic since titles do not provide sufficient evidence to discard publications. The entire selection process of this review would need to be revised as a result.

 The title and abstract were screened at the same time- with the title first, then the abstract- text corrected. 

c. Overall observation - Lack of clarity about outcome measures: Generally speaking, the authors declare they would like to assess the impacts or effectiveness of paediatric interventions for SCH control / elimination. However, at no moment they specify what they mean by "impacts / effectiveness". There is no operationalisation of outcome measures and hence there is no clarify in any section of the paper of what they are trying to assess. This is profoundly problematic for the overall structure of the review.

 Revised and mentioned in Lines 85 to 90

The outcome of the review to determine the reduction and elimination of schistomiasis among the paediatric population in sub-Saharan Africa. The intervention is the different control programmes which include MDA/ preventive chemotherapy, WASH, snail control, health education and behaviour modification. Hence the effectiveness of control programs to reduce and eliminate schistosomias in the peaditric population. 

RESULTS:

a. PRISMA chart inconsistency: The PRISMA chart indicates that there were 198 studies selected for full assessment and that 158 were excluded. If that's the case, the review would end up discussing 40 publications, not 30 as stated in the text. The consistency of the charts needs to be reviewed and amended.

 PRISMA flow diagram revised

168 studies were excluded for not meeting the study inclusion criteria. The predominant reasons for the exclusion of the studies were due to the reports on praziquantel efficacy (n=84), co-infection with soil-transmitted helminthiasis and malaria (n=43) and no interventions included (n=41)

30 articled were critically appraised and included in the study.

b. Graphs: There is no clarity as to why all graphs provide country-based comparisons / descriptions if the results section does not proeed to assess such differences. It is unclear what information they are trying to provide and how this is relevant to the overall study. The section was revised to give a better understanding of the findings

c. Overall observation - Results section fails to actually ellaborate on the results: The section includes substantive opinion and discussion. Actual references to the studies reviewed, beyond what is contained in the summary table, is minimal. This is not standard for a results section that demands authors to focus on expanding their findings from their review. The various references to opninion pieces, recommendations, or policy / technical guidelines should be moved to the discussion section. The entire results section should be revised in this regard. Thanks for the input. The results section has been revised accordingly.

DISCUSSION:

a. Overall concern: I am afraid that the discussion section is, as it stands, not fit for purpose. Since the results section has failed to provide any meaningful summary of key messages and since the overall assessment has not specific any specific outcomes measures to be reviewed, there are many uncertainties with regards to what is the core message that is being elaborated and what pieces of evidence may support it. Some examples of problematic assertions guiding the discussion section can be found below:

- Lines 365-367: "The review showed that despite preventive chemotherapy lowering schistosomiasis prevalence, chances of re-infection remained high in endemic areas" This is not entirely accurate. Re-infection is mentioned in the table but not in the core results narrative. In addition, there's no clarity as to how the risk of reinfection was assessed in this review. Some references sometimes only mention re-infection without providing any specific indicators.

- Lines 379-382: "Preventive chemotherapy must be combined with other schistosomiasis control programmes such as improved sanitation, hygiene, snail control, behaviour change and positive attitudes and practices that can

collectively achieve the end of transmission." Generally, I agree with the sentiment. However, the evidence provided in the review neither supports such a strong assertion nor it is really referred to in the results section. Note, morever, that the Zanzibar studies that you consider support this statement in fact do not provide any conclusive evidence of the kind. This is either a misinterpretation of the results of such studies or a misuse of such references.

- Lines 389-392: "The review indicates that including vector control in schistosomiasis control programmes will definitely lead to end of transmission especially in areas where there is high prevalence and noncompliance to MDA" Same as previous paragraph, the body of evidence to back such a strong claim is not presented in the text.

 Discussion section was also revised relative to the the results section

b. One final comment is that, as mentioned in the introduction section, there is a lack of clarity of what the contribution of the review is. The authors should make an effort to assess how and to what extent their results differ or expand on existing overall recommendations for elimination.

 Thank you, we have added a contribution (see conclusion section)

Reviewer #3: 

Dear authors,

Certainly, the article is extremely important for the epidemiological context of schoolchildren in Africa, since schistosomiasis is still an important public health problem. However, some considerations should be made in the text to make it clearer.

Introduction

1) As the authors robustly described the infection in children, I suggest making it clearer in the introduction what the upper limit of the age range of schoolchildren. Addressed line 59

2) Line 63-64 – authors could consider exploring what environmental improvements should be implemented as prevention for reinfection. For example, WASH interventions.

 Addressed line 66

Materials and method

A few methodological points for the authors to consider:

3) It is appreciable and important that the authors took care to register the systematic review in the PROSPERO database. However, it would be very important to describe some key information in methodological writing for readers, such as the guiding question of the work and the temporality or years considered.

 Research question:

Addressed Line 96-97

Years considered:

Addressed Line 100-101

4) The authors mention that several terms were used in the research, however, in the main text it would be more useful to leave the details to the supplementary file outline and cite the supplementary file in the text. Describe the groups of terms used: for example, the following search terms was used: schistosomiasis OR agents AND control strategies. And write a sentence about which control strategies were used. Were no secondary terms used in the search to target countries of interest? Most of the terms were removed and added in a supplementary file.

Secondary terms were used, as shown in the supplementary file. There were no countries of interest, except any country in sub-Saharan Africa that conducted paediatric schistosomiasis control programmes

Control programmes mentioned on line 154-155

5) Inclusion criterion 3: please, consider explaining better what types of case study/series? Was a clinical case report article considered? Or was it considered a case report related to a specific community, neighbourhood, or city? Since the authors talk about the impact of control programs, it would be of interest to also include information in the main text of the methodology, which study designs were considered. Study designs considered in this review addressed in line 116-118

6) In the description of data extraction, there was no behavioral information. Please consider describing that the information extracted considered behavioral data related to the disease. Addressed as behavioural information was explained in the summary of findings table under health education control programmes

7) Line 120-126: please, consider mentioning that the inclusion and exclusion criteria defined for the study were observed in all phases.

 Addressed Line 147-148

8) Line 135-136: evaluate and place this information in the “results” section. Moved to Results Line 165

9) Line 139-144: this information refers to the “selection process” section. Moved to the selection process section Line 122-129

10) Line 149: was only intervention studies considered? Observational with comparison groups? Study designs considered in this review addressed in line 116-118

11) Line 153: were interventions at the household level not considered? Please, justify. Community based interventions were considered, one study was considered that assessed intervention at household level. 

See summary of findings table

Results

12) Fig1.: please, consider describing the number of articles for each of the exclusion reasons (n=3257). Figure 1 addressed to schow many control programmes have been done outside the sub-Saharan regions of Africa and most studies were adult studies the focus of this review were for the paediatric population and studies with coinfections with other neglected tropical diseaseswere excluded as well

13) In the extracted data, there was no mention of extracting data on behavioural changes. Extracted under effects of health education

Addressed Line 325-340

14) Consider describing the publication dates of the studies in one sentence and the follow-up date of the study control program; The review process and the temporal effects of a disease control program are crucial.

 Addressed Line 229-231

15) Please, consider better describing the scenario before and after implementing program control actions. The title of the article presupposes an assessment of “impact” on infection rates, which leads us to understand that there would be “before” and “after” evaluation in relation to the application of the program/strategy. Maybe use “effect” instead of “impact”.

 The entire results section was revised 

16) Consider using the description of strategies (text after table 1) to describe with more focus what was used in the included studies. For example, the included study (23) used behavioural change strategies, but in the text of lines 283-302 there is a more general description. Addressed Line 325-340

17) There was no more detailed description of the methodological section in the “Critical Appraisal and Risk of Bias” section. Please, consider including in the text or supplementary material with indication in the text. Addressed in the supplementary material 

Discussion

18) Line362: the authors use the term “effects” and not “impact”. Please standardize and consider revising the term in the article title. Corrected to effects

19) The authors make a very satisfactory and important discussion from the point of view of the findings. The article is important because it is the first to address the topic in the region and an integrated paediatric strategy.

 Thank you

20) Recommendations (Lines 428-438): please, consider including recommendations for all features described in Line 150. Addressed

---

## [Decision Letter · Decision Letter 1]

23 Jan 2024

PONE-D-23-22834R1Effects of paediatric schistosomiasis control programmes in sub-Saharan Africa: A systematic reviewPLOS ONE

Dear Dr. Vere,

Thank you for submitting your manuscript to PLOS ONE. After careful consideration, we feel that it has merit but does not fully meet PLOS ONE’s publication criteria as it currently stands. Therefore, we invite you to submit a revised version of the manuscript that addresses the points raised during the review process. One reviewer still has some inquiries for you, please adress these questions accordingly prior for acceptance of your manuscript. 

We look forward to receiving your revised manuscript.

Kind regards,

Raquel Inocencio da Luz, Phd

Academic Editor

PLOS ONE

Journal Requirements:

**Additional Editor Comments:**

Dear Authors,

Thank you for adressing the questions to the reviewers.

there are still a few inquiries before acceptance of the manuscript,

if you are able to repond to the reviewers comments

Best regards,

Reviewers' comments:

Reviewer's Responses to Questions

**Comments to the Author**

1. If the authors have adequately addressed your comments raised in a previous round of review and you feel that this manuscript is now acceptable for publication, you may indicate that here to bypass the “Comments to the Author” section, enter your conflict of interest statement in the “Confidential to Editor” section, and submit your "Accept" recommendation.

Reviewer #1: All comments have been addressed

Reviewer #3: (No Response)

2. Is the manuscript technically sound, and do the data support the conclusions?

Reviewer #1: Yes

Reviewer #3: Yes

3. Has the statistical analysis been performed appropriately and rigorously? 

Reviewer #1: N/A

Reviewer #3: N/A

4. Have the authors made all data underlying the findings in their manuscript fully available?

Reviewer #1: Yes

Reviewer #3: Yes

5. Is the manuscript presented in an intelligible fashion and written in standard English?

Reviewer #1: Yes

Reviewer #3: Yes

6. Review Comments to the Author

Reviewer #1: My comments were fully addressed.

1. The countries were studies were done were included in lines 211-214 as follows: '.... the review2

includes studies from, Eastern(Kenya, Tanzania and Ethiopia), Western (Nigeria, Senegal,

Togo, Cote’ d’Ivoire and Niger), Southern (Zimbabwe and Madagascar) and Northern Africa

(Sudan).

Reviewer #3: (No Response)

7. PLOS authors have the option to publish the peer review history of their article (what does this mean?). If published, this will include your full peer review and any attached files.

Reviewer #1: No

Reviewer #3: No

---

## [Author Response · Author response to Decision Letter 1]

17 Feb 2024

Response to reviewers.

Comment

2) Line 66 -- I also suggest mentioning interventions to improve access to water and infrastructure for its use (such as piped water, tap water and other domestic structures), as these are those most associated with schistosomiasis.

Response

Line 84- 89

Added: Limiting water contact is essential in preventing the spread of schistosomiasis. Transmission is greatly reduced by interventions such as introducing washing sinks, swimming pools, and providing domestic water supplies such as piped water, tap water and fences along water bodies. Although in some settings activities like fishing, sand harvesting and agriculture could still expose people to infections. 

Comment

6) Ok, behavioral information was explained in the results summary table. But it needs to be indicated that it was information extracted from the articles by the authors. Please add the extracted data to the list. Item 5, line 155.

Response

Line 230

Added behavior change as information extracted from articles

Comment

12) Ok. But how many articles were excluded because they dealt with control programs outside Sub-Saharan Africa? How many articles were excluded because they dealt with adults? How many excluded because they were treating other diseases? How many due to the effectiveness of prazinquentel? This information is important for the replicability of the systematic review and recommended by the PRISMA method. At least some of these could be contained in the flowchart in Figure 1.

Response

Line 176

Limiters used during the search included: 

1. Age

2. Countries outside sub-Saharan Africa 

3. Systematic reviews 

Number of articles excluded and reasons for exclusion are indicated in figure 1. as:

1. Co-infections with STH and malaria-43

2. Reporting on PZQ efficacy-84

3. No intervention included -41

---

## [Editor Report · Decision Letter 2]

18 Mar 2024

Effects of paediatric schistosomiasis control programmes in sub-Saharan Africa: A systematic review

PONE-D-23-22834R2

Dear Author,

We’re pleased to inform you that your manuscript has been judged scientifically suitable for publication and will be formally accepted for publication once it meets all outstanding technical requirements.

Kind regards,

Raquel Inocencio da Luz, Phd

Academic Editor

PLOS ONE